# A 368-year maximum temperature reconstruction based on tree ring data in northwest Sichuan Plateau (NWSP), China

Liangjun Zhu[1], Yuandong Zhang[2], Zongshan Li[3], Binde Guo[1], Xiaochun Wang[1]

[1] Center for Ecological Research, Northeast Forestry University, Harbin 150040, China

[2] Key Laboratory of Forest Ecology and Environment, State Forestry Administration, Institute of Forest Ecology, Environment and Protection, Chinese Academy of Forestry, Beijing 100091, China

[3] State Key Laboratory of Urban and Regional Ecology, Research Center for Eco-Environmental Science, Chinese Academy of Sciences, Beijing 100085, China

*Correspondence to*: Xiaochun Wang (wangxc-cf@nefu.edu.cn), Yuandong Zhang (zydxju@163.com)

**Abstract.** We present a reconstruction of July-August mean maximum temperature variability based on a chronology of tree-ring widths over the period 1646-2013 AD in northern Northwest Sichuan Plateau (NWSP), China. A regression model explains 37.1 % of the variance of July–August mean maximum temperature during the calibration period from 1954 to 2012. Compared with nearby temperature reconstructions and gridded land surface temperature dates, our temperature reconstruction had high spatial representativeness. Seven major cold periods were identified including 1708–1711, 1765–1769, 1818–1821, 1824–1828, 1832–1836, 1839-1842 and 1869–1877, and three major warm periods occurred in 1655–1668, 1719–1730 and 1858–1859 from this reconstruction. The 18th and 20th centuries were warm with a lower frequency of low temperature events, while the 17th and 19th century were cold with a higher frequency of extreme low temperature events. The typical Little Ice Age climate can also be well represented in our reconstruction and obviously ended with climatic amelioration at the late of the 19th century. Moreover, the 20th century rapid warming wasn't obvious in the NWSP mean maximum temperature reconstruction, which implied that mean maximum temperature might play an important and different role in global change as unique temperature indicators. Multi-taper method (MTM) spectral analysis revealed significant periodicities of 170-, 49-114-, 25–32-, 5.7-, 4.6–4.7-, 3.0–3.1-, 2.5- and 2.1–2.3-year quasi-cycles at a 95% confidence level in our reconstruction. Overall, the mean maximum temperature variability in NWSP may be associated with global land-sea atmospheric circulation (e.g. ENSO, PDO or AMO) as well as solar and volcanic forcing.

**Keywords** Temperature reconstruction; West Sichuan Plateau; Tree-rings; *Picea purpurea*; solar and volcanic forcing; Little Ice Age; Climate change;

## 1 Introduction

Northwest Sichuan Plateau (NWSP), located between the body of Tibetan Plateau and Sichuan Basin, is the transition zone of Tibetan Plateau and the second step of China, which is one of the most sensitive and vulnerable areas to climate change (IPCC,

2013). Frequent temperature-related natural disasters, especially in summer (e.g., hailstones, frost damage), have a major influence on the natural ecosystem and human activities over the entire NWSP areas (Zheng and Yao, 2004;Ma, 2006). The climate in NWSP is complexly influenced by large-scale atmospheric circulation patterns, such as Asian monsoon and El-Niño/Southern Oscillation (ENSO), Atlantic Multidecadal Oscillation (AMO) and Pacific Decadal Oscillation (PDO) (Shao and Fan, 1999;Wu et al., 2005;Song et al., 2007;Qin et al., 2008;Duan et al., 2010;Yu et al., 2012;Xiao et al., 2013;Deng et al., 2014;Xiao et al., 2015), but driving mechanisms are not fully known or understood.

Long temperature records in NWSP are particularly critical for us to understand variability of past temperature variations and influences of large-scale atmospheric circulations as well as possible forcing mechanisms in order to predict future climate (Liang et al., 2008;Li et al., 2011;Thapa et al., 2014). Therefore, it is essential to explore long-term temperature variability from model-validated climate proxy data. However, most instrumental meteorological records in NWSP are of short length, covering only the past 40–60 years, and thus provide little information on climate variability over decades and longer.

Natural proxies, such as ice cores, speleothems, pollens, lake sediments, tree rings and so on, have the potential to fulfill this task (IPCC, 2013). Tree-ring data, a major proxy for paleoclimate research, have been widely used to reconstruct past climate worldwide (Briffa et al., 1995;D'Arrigo et al., 1998;Bräuning and Mantwill, 2004;Gou et al., 2008;Deng et al., 2014;Thapa et al., 2014;Wang et al., 2015) due to accurate dating, wide distribution, high resolution, and good replication (Stokes and Smiley, 1968;Fritts, 1976). In recent years, rapid progress in dendroclimatic reconstructions has occurred throughout China (Zhang, 2015). Among these reconstructions, only a few occurred in NWSP. Unfortunately, most of them were reconstructed temperature in a single month in summer or continuous months in winter, and measured mean or minimum temperature rather than mean maximum temperature (Shao and Fan, 1999;Wu et al., 2005;Song et al., 2007;Qin et al., 2008;Duan et al., 2010;Yu et al., 2012;Xiao et al., 2013;Deng et al., 2014;Xiao et al., 2015).

Global warming has already been regarded as an indisputable fact (IPCC, 2013). Recent studies suggest that global warming has occurred mainly in minimum temperatures and at night in most regions of Northern Hemisphere, however, evidence of warming trends of maximum temperatures were not evident; variations of the mean, minimum, and maximum temperatures were asymmetric during the warming processes. It is suggested that we should focus more attention on reconstructing various temperature variables (e.g. maximum and minimum temperatures, etc.) but not mean temperature (Wilson and Luckman, 2002;Wilson and Luckman, 2003). Therefore, the aims of the present study are to (1) develop new tree-ring width chronologies for the timberline forests in NWSP; (2) reconstruct past mean maximum temperature variations in late summer using the tree-ring chronology; (3) analyze and compare this reconstruction with the existing reconstruction and historical data archives in NWSP; and (4) identify possible driving mechanisms of late summer temperature in this area.

## 2 Materials and methods

### 2.1 Study area

Samples were collected from spruce (*Picea purpurea*) growing at timberline of Ayila mountain in Chali township, located in NWSP (32°43′49″ N, 102°06′17″ E; 3900 m above sea level (a.s.l.)). This region is classified as a sub-humid climate, and
strongly influenced by Asian monsoon, with dry periods from November to April and wet periods from May to October (Fig. 1). Based on records from the two nearest weather stations in Aba (1955-1990) and Hongyuan (1962-2013), the mean annual temperatures are 3.4 ℃ and 1.5 ℃, the mean monthly minimum temperatures are -5.4 ℃ and −2.9 ℃, and the mean monthly maximum temperatures are 10.2 ℃ and 12.2 ℃ (Table 1). The regional average annual precipitation ranges from 500 to 1000 mm, is highly variable on inter-annual timescales, and falls predominantly during the summer monsoon season (May to
October, 86%-89% ; Fig. 1).

The dark coniferous forests in this area are distributed within narrow bands on both sides of valley and generally reach an altitude of 3500–4,000 m a.s.l. *P. purpurea* is the most widespread and dominant species with 80% coverage, mainly growing on typical mountain brown dark coniferous forest soils (according to the Chinese Soil Taxonomic System) (Yang et al., 1992;Li, 1995). As a shallow-root tree specie, *P. purpurea* is able to endure shade and humid conditions in addition to extreme cold, so
that it usually grows at high altitudes where only a thin soil layer is covering the bedrock. The vegetation of this area is sub-alpine dark coniferous forests dominated by spruce, fir and cypress forests as well as some pines, birch and oaks, at low altitudes (Yang et al., 1992).

### 2.2 Tree-ring data

In July 2014, a total of 36 increment cores from 16 live trees were collected on opposite sides using an 5-mm diameter
increment at breast height (above ground level 1.3 m) to avoid or reduce errors in different sides caused by topography, competition, climate and growth characteristic. In order to remove the influence of identifiable stand disturbances (including animal and human disturbance, windstorm, snow and fire damage) and any obvious abnormal growth, each sample tree and sampling area were selected carefully. According to traditional dendrochronological procedures (Fritts, 1976;Cook and Kairiukstis, 1990), all cores were fixed by trogue and air dried at room temperature for 2–3 days in laboratory. Afterwards, we
progressively sanded those cores to a fine polish until individual tracheids within annual rings were visible. Then, cross dating was performed to assign calendar years to each growth ring and identify possible false or absent rings using a Skeleton-plot cross-dating method (Stokes and Smiley, 1968), and subsequently tree-ring width was measured at 0.001 mm resolution using a semi-automatic Velmex tree ring measurement system (Velmex, Inc., Bloomfield, NY, USA). Cross-dating and measurement accuracy were checked both first visually and then statistically using the COFECHA computer program (Holmes, 1983). To
remove the non-climate signals related to tree age or the effects of stand dynamics, cross-dated ring widths were detrended and standardized by fitting a conservative negative exponential curve or linear curve with negative slope using the ARSTAN program (Cook, 1985). We calculated ring width indices for each specimen by dividing the ring-width value by the value of

the fitted curve for each year. The resulting ring-width indices were averaged together to generate a standard STD chronology for collection site (Cook, 1985) (Fig. 2).

The quality of composite tree-ring chronology was evaluated by estimating the mean correlation between series (RBAR) and expressed population signal (EPS). The RBAR provides an indication of chronology signal strength (common variance between all series) and is independent of sample size (Wigley et al., 1984), whereas the EPS assesses the degree to which the chronology represents a hypothetical chronology based on an infinite number of cores and is calculated from between-tree correlation and number of trees included in the calculation (Briffa et al., 1995). Running RBAR and EPS statistics were calculated for 51- year intervals of the chronology with 25-year overlaps to assess the stability of signal strength as chronology replication diminished back in time. EPS has an overall mean of 0.90, well above the generally accepted 0.85 cut off (Wigley et al., 1984) except for a brief period in the 1690s when it falls to just above 0.8 (the Rbar decline to 0.37). The lower EPS and Rbar values Ca.1695 (Fig. 2) seem to result from suppressed growth during this cool period in more mature trees and somewhat erratic juvenile growth in the trees entering the chronology about this time (D'Arrigo et al., 2001). Although the EPS remains >0.85 early in the record (Ca. 1620), which might be caused by the inflated Rbar duo to a higher percentage of correlations being computed from within the same trees. To extend the length of the chronology, we thus consider this chronology to be most reliable over the past 368 years (AD 1646-2013), which corresponds to a minimum sample depth of 5 cores (four trees) for our chronology (Fig. 2).

**2.3 Climate and statistical analyses**

Instrumental climatic data in this study, provided by the China Meteorological Data Sharing Service System (http://cdc.cma.gov.cn/), were available from weather stations of Hongyuan (32°48′ N, 102°33′ E; 3248 m a.s.l.) and Aba (32°54′ N, 1101°41′ E; 3254 m a.s.l.). They were two nearest stations to tree-ring sampling sites of Chali with a maximum distance of about 42 km (Table 1). With similar seasonal rainfall distribution and climatic conditions, Hongyuan and Aba meteorological stations cover the period 1961–2013 and 1955–1990 of instrumental data, respectively (Fig. 1). Large-scale climate data (Atlantic Multidecadal Oscillation, AMO; Pacific Decadal Oscillation, PDO; Multivariate ENSO Index, MEI) were downloaded from the KNMI climate explorer (http://climexp.knmi/nl).

In order to determine the relationship between tree growth and climate factors, the initial observed climatic data (total precipitations, mean, minimum, and maximum temperatures) from the previous July to current September of the two stations were used to perform a correlation analysis. Based on stronger relationship between tree-ring index and previous July to August mean maximum temperature of Hongyuan and Aba, we reconstructed regional late summer (from July to August) mean maximum temperature (hereafter RLST) using a simple linear regression model. In order to obtain a long calibration/verification period of July-August mean maximum temperatures, the July and August mean maximum temperature in two stations were extended to the period 1955-2013 using linear regression of records from Hongyuan and Aba stations before reconstruction. Models explain 87 % of the variance in mean maximum temperature of July (correlation 0.93) and 92 %

of the variance of August (correlation 0.96), which indicates that variations in temperature with time during the overlapping period have been very similar throughout the region.

A traditional split-period calibration/verification method was used to explore the temporal stability and reliability of the reconstruction model (Fritts, 1976;Cook and Kairiukstis, 1990). Multiple statistical parameters, including Pearson's correlation coefficient ($r$), the $R$ square ($R^2$), the Sign test (ST), the reduction of error (RE), coefficient of efficiency (CE), the product means test (PMT), the Durbin-Watson (DW) and the root mean square error (RMSE), were used to evaluate model ability of this method (Fritts, 1976;Cook and Kairiukstis, 1990). To examine the temporal and spatial representativeness of the temperature reconstruction, spatial correlation between actual and reconstructed RLST variables in this study and gridded (0.5 °×0.5 °) temperature data from 1955 to 2012 (CRU TS3.22 Maximum Temperature, Harris et al., 2014) were performed using the KNMI climate explorer (http://climexp.knmi/nl). We also compared our RLST reconstruction for the past 368 years with other tree-ring-based temperature reconstructions from nearby areas and over a large scale. We further filtered those reconstructions with 11- and 61-yr moving averaged, and discussed the similarities among them in terms of decadal to multidecadal variations. Moving correlations were used to visualize and describe the periods of agreement and disagreement.

A spectral analysis was performed to identify the periodicity of local climate variability reconstructed in this study using Multi-taper method (MTM) program (Mann and Lees, 1996). Teleconnections between reconstructed RLST variables and monthly Optimum Interpolation Sea Surface Temperature (0.5 °×0.5 °, monthly OISST) were also performed to verify the potential driving of large-scale climate on local temperatures. The NOAA monthly OISST is an analysis constructed by combining observations from different platforms (satellites, ships, buoys) on a regular global grid. Although the monthly OISST data is short (1982-now) but more accurate. Therefore, we chose it instead of other SSTs data.

Superposed epoch analysis (SEA) was used to test the presence of a volcanic cooling signal in our reconstruction (Haurwitz and Brier, 1981). In this case, an 11-yr window was used, 6 yr before and 4 yr after the event years. Departures from the mean temperature values for the key years and for windows of years were superposed and averaged to determine if temperature for these years was significantly different from randomly selected sets of 11 other years (Fischer et al., 2007). A standard t test and a Monte Carlo method were used to test the significance of the means, and 10000 simulations were performed by random resampling to determine the probability associated with the average departures for the volcano key dates (Haurwitz and Brier, 1981). SEA was conducted using software FHAES version 2.0.0 (https://www.frames.gov/partner-sites/fhaes/download-fhaes/). According to Davi et al. (2015), a large tropical volcanic event year list (1601, 1642, 1763, 1810, 1816, 1836, 1992) generated from Gao et al. (2008) was used to estimate of global volcanic forcing during the years 1554 – 2012. These years were chosen based on querying the global forcing series for years with negative forcing of at least the magnitude of the Pinatubo (1991) eruption, and then using only the first year if there were multiple consecutive years with large negative forcing.

# 3 Results

## 3.1 Tree growth–climate relationships

We used simple correlation analysis to identify climate signals in ring-width chronology through SPSS software (IBM, Armonk, New York). Relationships between Chali chronology and Hongyuan and Aba monthly climate data were shown in Figure 3. A significant positive correlation ($p < 0.05$) between tree radial-growth of *P. purpurea* and mean and maximum temperature in the previous July and August was found at both of the weather stations, which indicated that the late summer mean maximum temperature in the previous year played a vital role for radial growth of *P. purpurea*. Ring-width chronology of *P. purpurea* was negatively ($p < 0.05$) associated with previous November mean and minimum temperature at the Hongyuan weather station, and a similar negative correlation relationship was also found at the Aba weather station, though not at a significant level. There was a significant negative correlation ($p < 0.05$) between tree radial-growth of *P. purpurea* and mean minimum temperature in June found at the Hongyuan weather station, however it was not significant at the Aba weather stations. The correlations between monthly precipitation data and the ring-width index were not significant at both weather stations. Therefore, we selected July-August mean maximum temperature as the target of our reconstruction.

## 3.2 Regional temperature reconstruction

We reconstructed RLST history based on the strong relationship between previous July and August mean maximum temperature and tree-ring index (TRI). A simple linear regression model between TRI and the composite July-August mean maximum temperature of Hongyuan and Aba weather stations covering form 1955 to 2013 was created to reconstruct regional temperature history. The model is as follows:

$$T_{JAt} = 5.35I_{t+1} + 13.36, \quad (r = 0.61, \text{ N} = 58, R^2 = 37.1\%, R^2_{adj} = 36.9\%, F = 32.97, P < 0.001) \tag{1}$$

where $T_{JAt}$ is RLST at year t and $I_{t+1}$ is the ring width index at year $t+1$. The RLST reconstruction model accounted for 37.1% (36.9% after adjusting for the degrees of freedom) of the composite mean maximum temperature variance for calibration period from 1955 to 2012 (Fig. 4A, Table 2). Shown in Fig. 4A, our temperature reconstruction couldn't fully capture the magnitude of extraordinary warm or cold years at high frequency especially in the last few years similar to other tree-ring reconstructions (D'Arrigo et al., 1998), but it paralleled the general tendency of composite RLST during calibration period. Spatial correlation analysis showed a similar general pattern between the observed and estimated RLST compared with gridded temperature records (Fig. 5). These results suggest that our reconstructions provide some information about late summer mean maximum temperature variability in NWSP.

Split-period calibration/verification analysis (Fritts, 1976;Cook and Kairiukstis, 1990) was used to test the stability and reliability of the regression model including statistical parameters of $r$, $R^2$, RE, CE, ST, PMT, DW and RMSE. Two rigorous tests of fit, the RE and CE were both strongly positive (Table 2) indicating that the model was significantly and considerably skillful in reconstructing observed variations (Fritts, 1976;Cook and Kairiukstis, 1990). The statistical parameters of $R^2$, ST and PMT all exceeded the 95% or 99% confidence level. The DW, used to analyse reconstruction residuals from a regression

analysis, ranged from 1.68 to 1.99, indicating no significant autocorrelation or linear trends among our residuals (Fig. 4B-C; Table 2). Moreover, the RMSE value was relatively small in our model (Cook and Pederson, 2011). Upon successful validation of the two split-period models, regression parameters for the full calibration period were used to reconstruct RLST back to AD 1645 based on the ring-width record.

## 3.3 Regional temperature variations

According to the regression equation, our reconstructed RLST could be extended back to A.D. 1645 with a mean of 18.75 ℃ and a standard deviation (SD) of ±0.84 ℃ (Fig. 4B). In this paper, we defined those having values (11-year moving average series) exceeding 19.59 ℃ (mean＋SD) as extremely warm periods, while those not exceeding 17.91℃ (mean－SD) as extremely cold periods. Relatively cold periods occurred during 1708–1711, 1765–1769, 1818–1821, 1824–1828, 1832–1836, 1839-1842 and 1869–1877. Warm periods prevailed during 1655-1668, 1719-1730 and 1858-1859. Among them, the ten coldest yeas were identified as 1872(16.49), 1764(16.55), 1766(16.68), 1874(16.94), 1707(17.01), 1932(17.05), 1680(17.1), 1770(17.1), 1705(17.15) and 1837(17.16), and the top ten warmest years were 1719(21.33), 1720(21.05), 1660(21.04), 1726(20.68), 1723(20.66), 1662(20.64), 1729(20.56), 1661(20.53),1911(20.52) and 1951(20.5), respectively.

During the period 1875–1955, late summer temperature fluctuated less strongly than before or thereafter. In general, the average length of cold periods was shorter than that of warm periods. The cold period of 1869-1877 was the longest and coldest cool period with a mean of 17.63 ℃. The longest warm period extended from 1655 to 1668, while the warmest period in AD 1719-1730 with a mean of 20.37 ℃. However, we should point out that the rapid warming during the 20[th] century wasn't extraordinary obvious in our reconstructed RLST.

A multitaper method (MTM) of spectral analysis (Mann and Lees, 1996) was performed to identify major periodicities present in the full range of our reconstruction. The result of spectral analysis over the full range (1645-2012) showed that there were significant periodicities at 2-2.3, 2.5, 3-3.1, 4.6-4.7, 5.6, 25-32 and 53-93 years at the 95% confidence level (Fig. 6).

## 4 Discussion

### 4.1 Climate-growth relationship

Temperature, especially winter or growing season temperature, limits tree growth at sub- and alpine treelines at high latitudes of Northern Hemisphere, as suggested by previous dendroclimatic and seasonal cambial activity studies (Körner and Paulsen, 2004;Rossi et al., 2008;Seo et al., 2013). Large temperature sensitivity, especially during the summer season, has also been clearly demonstrated in tree line sites of the eastern Tibetan Plateau (Fan et al., 2009;Zhu et al., 2011;Li et al., 2012). In our study, the radial growth of *P. purpurea* was significantly positive correlated to mean and maximum temperature in July and August of the previous year, which suggests that previous late summer temperature was the major limiting factor to tree-ring growth of *P. purpurea* in NWSP, China (Fig. 3). Similar results showed that the radial growth of *P. purpurea* was majorly

limited by late summer temperature of previous and current years (Bräuning and Mantwill, 2004;Ren et al., 2014;Guo et al., 2015), and had been reported by previous studies in nearby areas. Most of these have been used to reconstruct the historical temperature variations (Bräuning and Mantwill, 2004;Ren et al., 2014). Tree-ring chronologies were an indicator of late summer temperature, which was also found at high-elevation conifer sites on the eastern Tibetan Plateau (Bhattacharyya and Chaudhary, 2003;Bräuning and Mantwill, 2004), which further suggested that the spruce growth at timberline in this area was greatly affected by previous late summer temperature.

The lagged relationship to climate is common in dendroclimatology analyses (Bräuning and Mantwill, 2004;Gou et al., 2006;Duan et al., 2010;Zhu et al., 2011;Li et al., 2012), which usually involved the use of stored photosynthetic product during the current growing season and/or developmental processes (e.g. leaf maturation, root elongation) in the previous growing season (Fritts, 1976). It is possible that more non-structural carbohydrates (NSC) and other organic substances, which could be used to produce more wood in the next spring at high-elevation sites, were synthesized in a warmer late summer (Li et al., 2008). In contrast, by increasing the frequency of frost and missing rings and limiting the growth of roots and their function in water uptake (Körner and Paulsen, 2004), a narrow ring would usually be formed and less NSC and other organic substances would be synthesized in a cold late summer.

Tree growth at timberlines not only depended on growing season temperature, but also occasionally on winter temperatures which could cause frost drought in evergreen conifers (Oberhuber, 2004;Elliott, 2012). Generally, the radial growth of *P. purpurea* had a negative relationship with winter (from previous October to current April, especially in November) mean and minimum temperature rather than maximum temperature (Fig. 3). Warm winters often accompanied by low snow cover, which might particularly and likely lead trees to suffer from enhanced frost desiccation as a result of increased transpiration rates of needles and shoots, photoinhibitory stress and short-term fluctuations in shoot temperatures, leading to xylem embolism (Sperry and Robson, 2001). Sustained winter defoliation at timberline had an overwhelming influence on subsequent radial growth (Payette et al., 1996;Oberhuber, 2004). However, more snow protects plants against these stresses during the cold season (Sakai and Larcher, 1987). These explained negative correlation coefficients between mean minimum temperature in winter and *P. purpurea* growth at timberline. Therefore, significant positive correlations of the radial growth of *P. purpurea* with the previous late summer mean and maximum temperature and positive correlation of the radial growth of *P. purpurea* and the previous winter mean minimum temperature were reasonable and meaningful in this study, based on inferences from known physiological processes.

**4.2 Comparison with regional records**

To assess whether our reconstruction represents features and evaluate the spatial synchrony of temperature variation, a spatial correlation analysis was performed between the 0.5 °×0.5 °gridded July-August mean maximum temperature data (CRU TS3.22; Harris et al., 2014) for the period 1955-2012 over a large region in NWSP. Both the observed and reconstructed late summer temperature in NWSP showed a similar spatial correlation pattern with regional temperature records, which was coherent over a large spatial scale, including most of the Tibetan Plateau, central and eastern regions of China, and parts of Mongolia and

Inner Mongolia, China (Fig. 6). We also compared our reconstruction with other tree-ring based temperature reconstructions in surrounding regions (Fig. 7). The RLST variations in NWSP were coherent over a large spatial scale, also including most of the Tibetan Plateau, central and eastern regions of China, and parts of Mongolia and Inner Mongolia, China. July-August mean temperature, based on tree-ring data obtained nearby our study site, in the north of west Sichun of Xiao et al. (2015) was significantly positively ($r = 0.122$, $p = 0.028$) associated with our reconstruction (Fig. 7). Additionally, a significant positive correlation ($r = 0.1$, $p = 0.039$) between our reconstruction and February-June temperature reconstruction for Kathmandu (Cook et al., 2003), located farther from our study site, was also found (Fig. 7). Both the sites (farther and near our study site) displayed highly synchronous variation patterns with our reconstruction overall, which further validated results of spatial correlation analysis. Moving correlation analysis showed that our reconstruction temperatures were similar with Xiao et al. (2015)(except the period 1876-1975) and Cook et al. (2003) (except the periods 1676-1825 and 1876-1975) in decadal variations (11-yr moving averaged), and also shown consistency with Xiao et al. (2015) (except the period 1826-1925) and Cook et al. (2003) (except the period 1676-1825) in multidecadal variations (61-yr moving averaged) (Fig. 8). An opposite correlation pattern was found between Xiao et al. (2015) and Cook et al. (2003) (Fig. 8), significantly negative and positive/ positive and negative correlations between our reconstruction and the reconstructions of Xiao et al. (2015) and Cook et al. (2003) were identified in the year of 1676-1975 (Fig. 8). Such widespread temporal and spatial variability of temperature had been found in previous studies (Legates and Willmott, 1990;Yang, 2001). However, those decadal or multidecadal differences between them might be due to differences in season, species, as well as the different standardization approach applied on tree-ring data (Thapa et al., 2014). In addition, inherent differences in climate variables and in climate-driven mechanisms were also the main sources of differences. For example, variations of the mean, minimum, and maximum temperatures were often asymmetric (Wilson and Luckman, 2002;Wilson and Luckman, 2003;Gou et al., 2008). Asymmetric variability between mean, minimum and maximum temperatures was also found at the Hongyuan weather stations, especially the differences between mean minimum and maximum temperature.

Seven cold periods and three warm periods were identified during the past 368 years (Fig. 4D). All the cold periods were during the Maunder (1708-1711) or Dalton (1818–1821, 1824–1828, 1832–1836 and 1839–1842) solar minima periods except for the cold periods of 1765–1769 and 1869–1877 (Eddy, 1976;Shindell et al., 1999), which indicated that RLST variations in NSWP might be driven by solar activity (Fig. 7B). On the other hand, volcanic eruptions in the corresponding periods might also be a cooling factor (Fig. 7B). Longer cold period (e.g. 1820s-1840s) was interrupted by transient warming, thus forming a plurality of discontinuous short cold periods. Warm periods of 1719-1730 and 1858-1859 both had more sunspots (Eddy, 1976;Shindell et al., 1999) and lower volcanic forcing (Fig. 7B). The cold (1765–1769 or 1869–1877) and warm (1655-1668) periods were highly consistent with other studies (Fig. 7). It's hard to explain the cold (1765–1769 or 1869–1877)/warm (1655-1668) periods without/with obvious volcanic and solar forcing (Fig. 7B), but it might be linked with volcanic and solar activities (Haurwitz and Brier, 1981;Shindell et al., 1999;Fischer et al., 2007;Gao et al., 2008;Breitenmoser et al., 2013;Stoffel et al., 2015). It was noteworthy that 20[th] century warming (Song et al., 2007;Zhu et al., 2011;IPCC, 2013) was not very obvious in our reconstruction, which was in line with recent studies that global warming has occurred mainly in the minimum

temperatures and at night in most regions of the Northern Hemisphere, however, evidence of the warming trend of the maximum temperatures was not clear (Wilson and Luckman, 2002;Wilson and Luckman, 2003). Moreover, the Little Ice Age (LIA) climate can also be well represented, though the magnitude and timing of maximum glacierization varied regionally, especially in the vast and complex terrain of China (Yang, 2001). The LIA in our reconstruction obviously ended with climatic

amelioration at the end of the 19th century (Fig. 7B). In the nearly 400-year time scale, 18th and 20th centuries were mainly in warm with a lower frequency of low temperature events, while 17th and 19th century were cold with a higher frequency of extreme low temperature events (Fig. 7B), which were consistent with Yang (2001). In short, our regional reconstruction could be a good representative of the past late summer maximum temperature variations in NWSP.

## 4.3 Possible forcing mechanism

Results of MTM analysis revealed that some significant dominant periodicities existed in our reconstructed regional temperature variability (Fig. 6). Among them, significant high-frequency periodicities around 5.7, 4.6–4.7, 3.0–3.1, 2.5 and 2.1–2.3 years, falling within the range of El Niño Southern Oscillation (ENSO) cycle of 2-8 years (Holton et al., 1989), were found in our reconstruction, which indicated that ENSO might affect the late summer temperature variability in NWSP. Consistent with this observation, ENSO may have a strong influence on temperature variability in neighbouring

dendroclimatological research, such as northwest Yunnan (Fan et al., 2009;Li et al., 2011;Li et al., 2012), Tibetan Plateau (Bhattacharyya and Chaudhary, 2003;Bräuning and Mantwill, 2004;Gou et al., 2006;Gou et al., 2008;Zhu et al., 2011) and West Sichuan Plateau (Duan et al., 2010). Strong teleconnections of ENSO with interannual variability of tree growth and local temperature with tropical ocean-atmosphere systems had been found in nearby sites (Liang et al., 2008;Deng et al., 2014). Based on these results we performed correlation analysis of the smoothed (11-yr moving averaged) temperature series with

extended Multivariate ENSO Index (MEI, Wolter and Timlin, 2011). A significant negative correlation was found between estimated RLST and MEI from the previous May to current August (Table 3) revealing that the cool (warm) previous and current SSTs might have a significant impact on late summer temperature in NWSP. Similar results were detected in western Nepal Himalaya region (Thapa et al., 2014) and Northern China  (Lu, 2005). These high-frequency cycles around 2–3 year might also linked to the quasi-biennial oscillation (QBO, Labitzke and Van Loon, 1999) and tropospheric biennial oscillation

(TBO, Meehl, 1987), which may also suggest strong teleconnections of local temperature variability with tropical ocean-atmosphere systems. The significant cycle around 25–32 years might be linked to Pacific Decadal Oscillation (PDO, Mantua et al., 1997), and significantly negative correlations of our smoothed (11-yr moving averaged) temperature reconstruction with PDO index from the previous May to current August were also detected (Table 3). PDO signals had been found in tree rings in large-scale Asian regions (D'Arrigo and Wilson, 2006). Accompanied by significant peaks at 60.2- and 73-years, the

continuously periodicities around 49-114 years in our regional temperature reconstruction might tentatively be related to PDO, Atlantic Multidecadal Oscillation(AMO, Enfield et al., 2001) as well as solar activity (Eddy, 1976;Shindell et al., 1999;Peristykh and Damon, 2003;Raspopov et al., 2004;Braun et al., 2005). The AMO was an important driver of multidecadal variations in summer climate not only in North America and Western Europe  (Kerr, 2000;Sutton and Hodson, 2005), but also

in the East Asian (Feng and Hu, 2008;Liang et al., 2008;Wang et al., 2011;Zhu et al., 2011;Wang et al., 2015). The 60.2-year peak associated with AMO demonstrated that multidecadal variations of late summer temperature in NWSP might be controlled by AMO. This was also consistent with recent findings; the AMO could change the land-sea thermal contrast between the East Asian continents thus affecting temperature of the East Asian (Feng and Hu, 2008;Wang et al., 2011). Further,

correlation analysis between the smoothed (11-yr moving averaged) temperature reconstruction and the monthly AMO index was calculated to test this hypothesis. A significant negative correlation during the previous July to September period and current May to August period was detected (Table 3), which further confirmed the powerful influence of AMO on late summer temperatures in northwest Sichuan. In addition, we also found significant cycles at more than 147 years with peaks at 170 years. Considering the limited length of our reconstruction (368 yrs), this cycle might not be reliable and must be interpreted

with caution, although there have been similar findings using temperature reconstruction in Eastern (Wang et al., 2015), north eastern (Liu et al., 2011) and south eastern (Deng et al., 2014) Tibetan Plateau. Strong teleconnections between reconstructed RLST variables and monthly July-August OISSTs of the Pacific, Indian and Atlantic Oceans were also found (Fig. 9). The RLST variables significantly correlated with the SSTs changes, especially the west and equatorial Pacific as well as the north Atlantic Oceans (Fig. 9), which further verified the driving of large scale climate (e.g. ENSO, PDO and AMO) on the RLST

in NWSP from another angle.

In addition to large-scale climate, solar and volcanic forcing might be also the key factors in driving the RLST variability in NWSP. Regional temperatures associated with solar activity in northwest Sichuan have been found (Shao and Fan, 1999;Xiao et al., 2013;Xiao et al., 2015), even the classic 11-yr sunspot cycle (Shindell et al., 1999) was not significant in our temperature reconstruction. However, those significant multidecadal and centennial-scale cycles of our temperature reconstruction might

include the signs of solar activity, such as the Gleissberg cycles (Peristykh and Damon, 2003), Suess cycles (Braun et al., 2005), Bruckner cycles (Raspopov et al., 2004) and Schwabe cycles (Braun et al., 2005). The Maunder (circa AD 1645–1715) and Dalton (circa AD 1790–1840) solar minima periods were generally associated with temperature depressions (Eddy, 1976), and the Damon (circa AD 1890–1920) solar maximum period occurred in a relatively warm period, which further confirmed that late summer temperature variation in NSWP might be driven by solar activity (Fig. 7B). In addition, volcanic cooling can

influence tree growth, such as the formation of frost-rings, light-rings as well as missing-rings, and provides accurate, impartial and coherent information on the timing and magnitude of such events and their influence on the climate system (Fischer et al., 2007;Gao et al., 2008;Breitenmoser et al., 2013; Stoffel et al., 2015). Cooling associated with explosive volcanic eruptions can also be found in our reconstruction of temperatures (Fig. 7B). The radiative forcing for ice core volcanic indices series after adjustments (Crowley, 2000) and our temperature reconstructions presents a high consistency (Fig. 7B). Most of volcanic eruptions

events lead a temperature drop in the year and subsequent years, especially in 1763 and 1810 (Fig. 10A). The result of SEA also shows significant cooling occurred during the year of the event (in Year 0) as well as the following years (in Year 1, 3) (Fig. 10B). Other dendroclimatological studies in the Tibetan Plateau, Central Asia, Europe as well as the Northern hemisphere also indicated an association between summer temperature reconstructions and volcanic eruptions (Fischer et al., 2007;Liang et al., 2008;Breitenmoser et al., 2013;Davi et al., 2015;Stoffel et al., 2015).

## 5 Conclusion

In this study, a high resolution tree-ring chronology was used to reconstruct RLST in NWSP from 1645 to 2012. The model of our reconstruction explains 37.1 % of the temperature variance. Spatial correlations with gridded land mean maximum temperature data and comparison with other nearby temperature reconstructions further revealed that the RLST has high spatial

representativeness, although there have some temporal and spatial differences in the low-frequency changes of temperature. Seven major cold periods and three major warm periods were identified from this reconstruction, which might be linked with volcanic and solar activities. The 18[th] and 20[th] centuries were warm with a lower frequency of low temperature events, while the 17[th] and 19[th] century were cold with a higher frequency of extreme low temperature events. Significant periodicities at 2-2.3, 2.5, 3-3.1, 4.6-4.7, 5.6, 25-32 and 53-93 years exist in our reconstruction. The 20[th] century rapid warming wasn't obvious

in our RLST reconstruction, which implies that mean maximum temperature, as a unique temperature indicator, might play an important and different role in global change. Moreover, the Little Ice Age (LIA) climate can also be well represented and obviously end with climatic amelioration at the end of the 19[th] century. Overall, the RLST variability in NWSP might be associated with global land-sea atmospheric circulation (e.g. ENSO, PDO or AMO) as well as solar and volcanic forcing.

**Acknowledgements**  This research was supported by the National Natural Science Foundation of China (Nos. 31370463 and 41471168), the Program for New Century Excellent Talents in University (NCET-12-0810), and the Program for Changjiang Scholars and Innovative Research Team in University (IRT-15R09). We would like to thank Alison Beamish at the University of British Columbia for her assistance with English language and grammatical editing of the manuscript. We also thank the staff of Aba Forestry Bureaus for their assistance in the field.

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

**Tables**

Table 1 Information of two weather stations sites nearest to sampling sites.

| Weather stations | Hongyuan | Aba |
|---|---|---|
| Long. (E) | 102°33′ | 101°41′ |
| Lat. (N) | 32°48′ | 32°54′ |
| Elev. (m) | 3248 | 3254 |
| P (mm) | 749.89 | 740.77 |
| T (℃) | 1.53 | 3.43 |
| T$max$ (℃) | 10.24 | 12.22 |
| T$min$ (℃) | -5.35 | -2.93 |
| Period | 1960-2013 | 1955-1990 |
| DWS (km) | 42.4 | 42.3 |

Note: P is annual total precipitation. T, T$max$ and T$min$ are annual mean, maximum and minimum temperature, respectively.

DWS is the distance from weather station to the sampling site of Chali.

Table 2 Calibration and verification statistics for the regional Jul-Aug mean maximum temperature reconstruction

| Calibration | $r$ | $R^2$ | Verification | $R^2$ | RE | CE | ST | PMT | DW | RMSE |
|---|---|---|---|---|---|---|---|---|---|---|
| 1955-2012 | 0.61** | 0.37** | — | — | 0.37 | — | (41,17)** | 6.46** | 1.68 | 0.73 |
| 1955-1984 | 0.78** | 0.61** | 1985-2012 | 0.28** | 0.26 | 0.21 | (21,8)* | 4.04** | 1.99 | 0.82 |
| 1985-2012 | 0.53** | 0.28** | 1955-1984 | 0.61** | 0.61 | 0.59 | (24,5)** | 4.63** | 1.69 | 0.62 |

$* = p < 0.05$, $** = p < 0.01$.

Table 3 Correlation of the smoothed (11-yr moving averaged) temperature reconstruction and other large-scale climate system cycles.

| Month | AMO (n=156) | PDO (n=112) | MEI (n=62) |
|---|---|---|---|
| PMAY | -0.15 | -0.43** | -0.43** |
| PJUN | -0.19* | -0.26** | -0.42** |
| PJUL | -0.19* | -0.27** | -0.32* |
| PAUG | -0.17* | -0.27** | -0.27* |
| PSEP | -0.17* | -0.24* | -0.27* |
| POCT | -0.07 | -0.19* | -0.29* |
| PNOV | -0.05 | -0.21* | -0.26* |
| PDEC | -0.07 | -0.21* | -0.28* |
| JAN | -0.00 | -0.34** | -0.28* |
| FEB | -0.13 | -0.29** | -0.27* |
| MAR | -0.13 | -0.39** | -0.38** |
| APR | -0.13 | -0.39** | -0.41** |
| MAY | -0.18* | -0.46** | -0.43** |
| JUN | -0.22** | -0.31** | -0.42** |
| JUL | -0.22** | -0.30** | -0.31* |
| AUG | -0.21** | -0.30** | -0.28* |

AMO: Atlantic Multidecadal Oscillation; PDO: Pacific Decadal Oscillation; MEI: Multivariate ENSO Index

* $p < 0.05$, ** $p < 0.01$

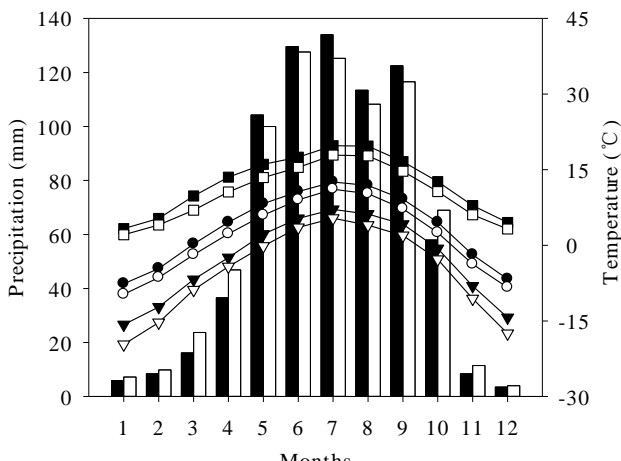

**Fig. 1** Monthly variation of total precipitations (bars), mean maximum temperature (line with squares), mean temperature (line with circles), and mean minimum temperature (line with triangles) in Hongyuan (filled with white) and Aba (filled with black) meteorological stations, calculated for the period of 1961–2013 and 1955–1990, respectively.

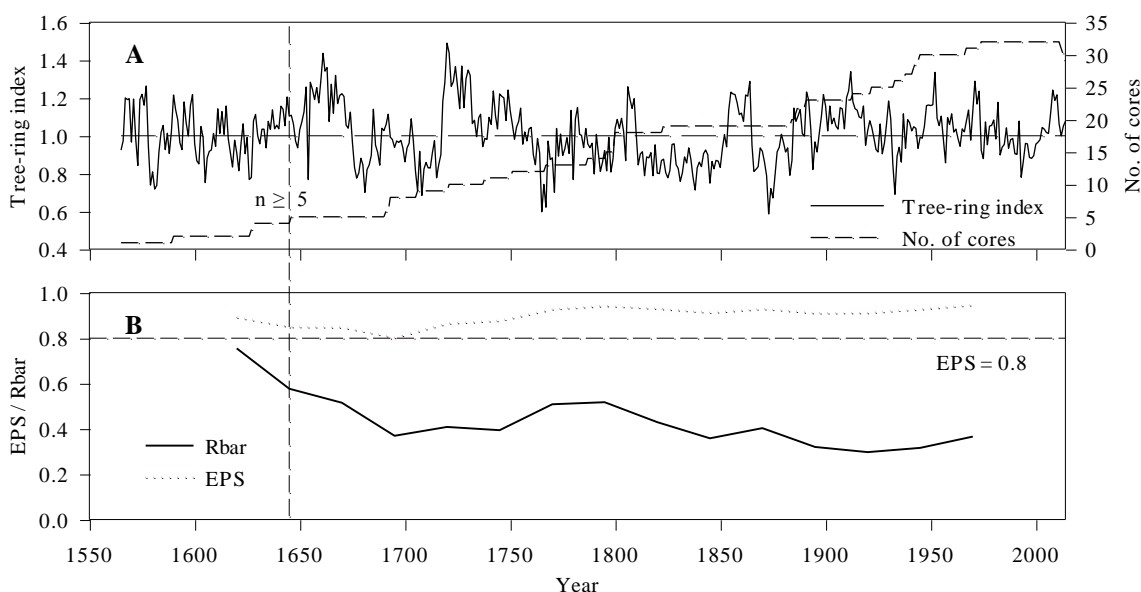

**Fig. 2** The tree-ring chronology of *Picea purpurea* at timberline in Chali. A, Standard chronology and sample size changing with time and B, Rbar and EPS (see text for explanation) plotted for 50-year windows with 25-year overlap for the standard

5   chronology. The *vertical dashed line* represents the cut-off point when the n ≥ 5 and EPS > 0.8. The *horizontal red dashed line* represents the theoretical 0.8 threshold.

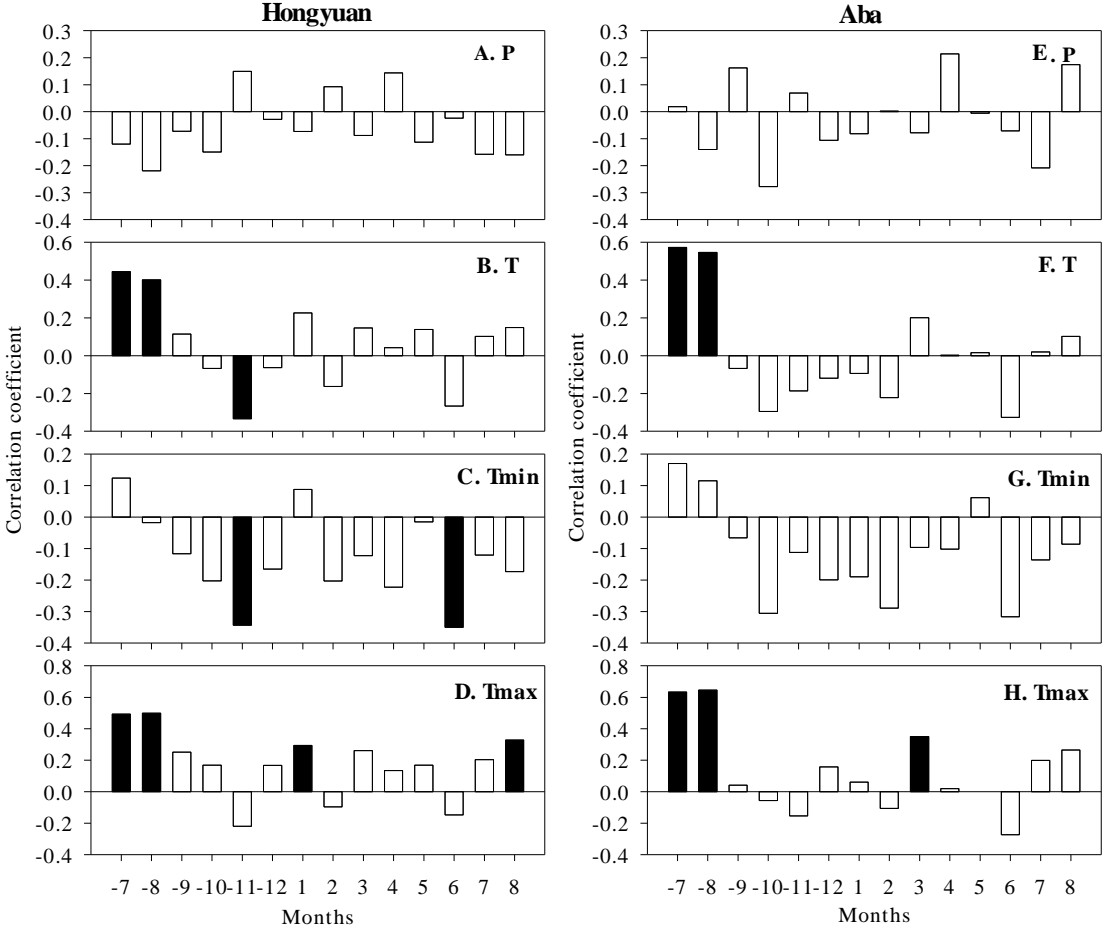

**Fig. 3** Correlation analyses between the Chali tree-ring chronology and meteorological climate data of Hongyuan and Aba including total precipitation (A, E), mean temperature (B, F), minimum temperature(C, G), and maximum temperature (D, H). The black-filled bars represent significant effects at 95 % significant levels.

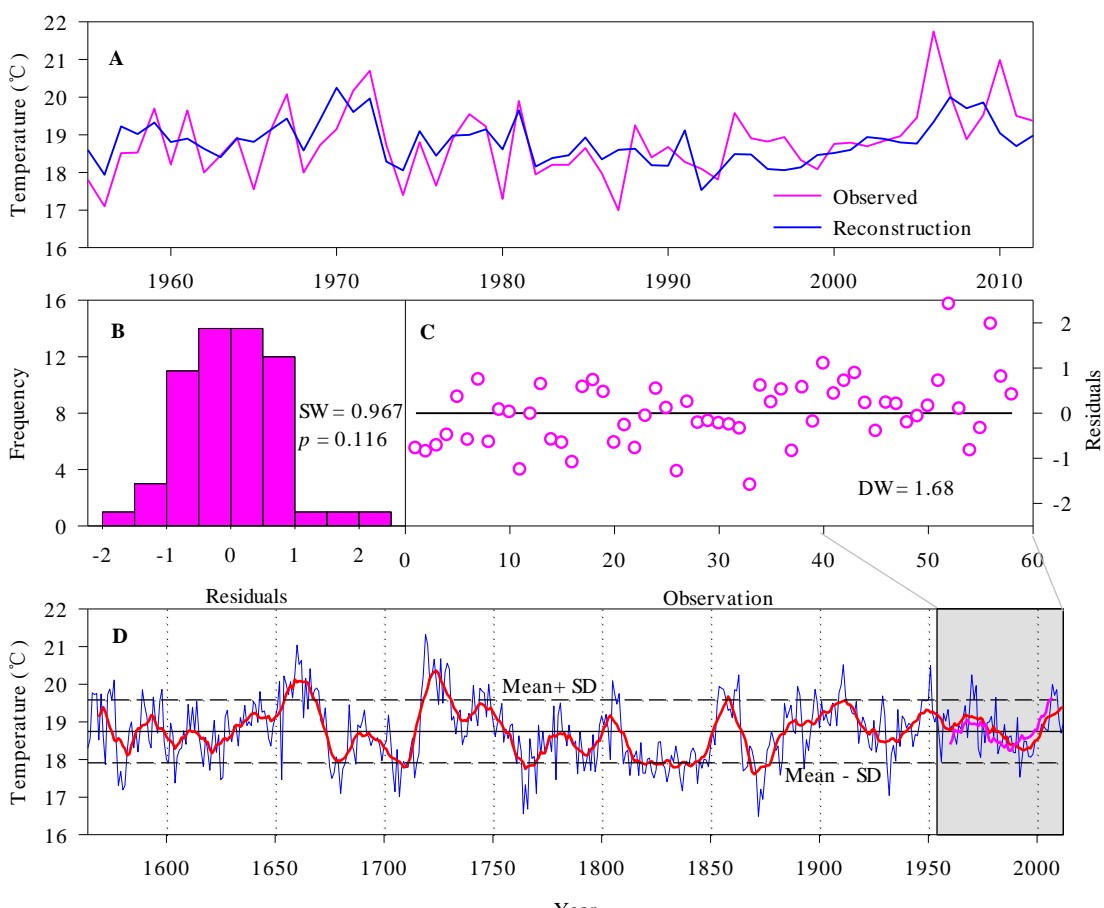

**Fig. 4** Temperature reconstruction in NWSP, China. A, The comparison of observed (pink line) and reconstructed (blue line) regional Jul-Aug mean maximum temperature during the calibration period 1955-2012; B, Shapiro–Wilk residuals normality test (SW); C, Durbin-Watson test for residuals autocorrelation (DW); D, Annual and 11-year smoothed regional Jul-Aug mean maximum temperature reconstruction (bule and tick red lines, respectively). Superimposed is the 11-year smoothed instrumental record (pink line).

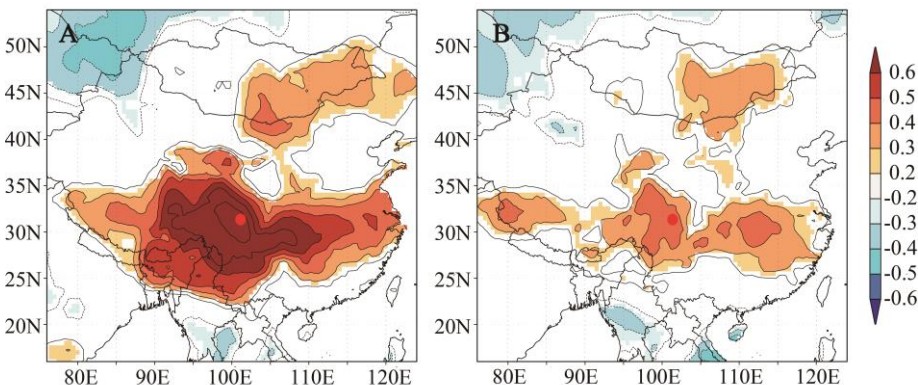

**Fig. 5** Spatial correlation fields of (A) instrumental and (B) reconstructed July-August mean maximum temperature for NWSP with regional gridded July-August mean maximum temperature for the period 1955–2011.

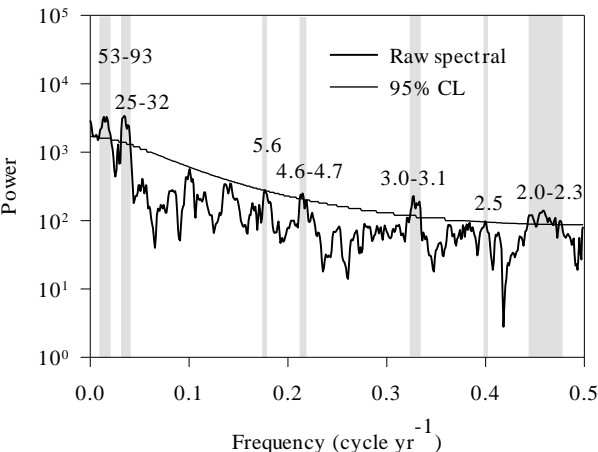

**Fig. 6** Multi-taper method power spectrum of the reconstructed July-August temperature for the period AD 1645-2012. The 95% confidence level relative to red noise is shown by the dashed curve. The gray area indicates the 95% significance level and the numbers refer to the significant period in years.

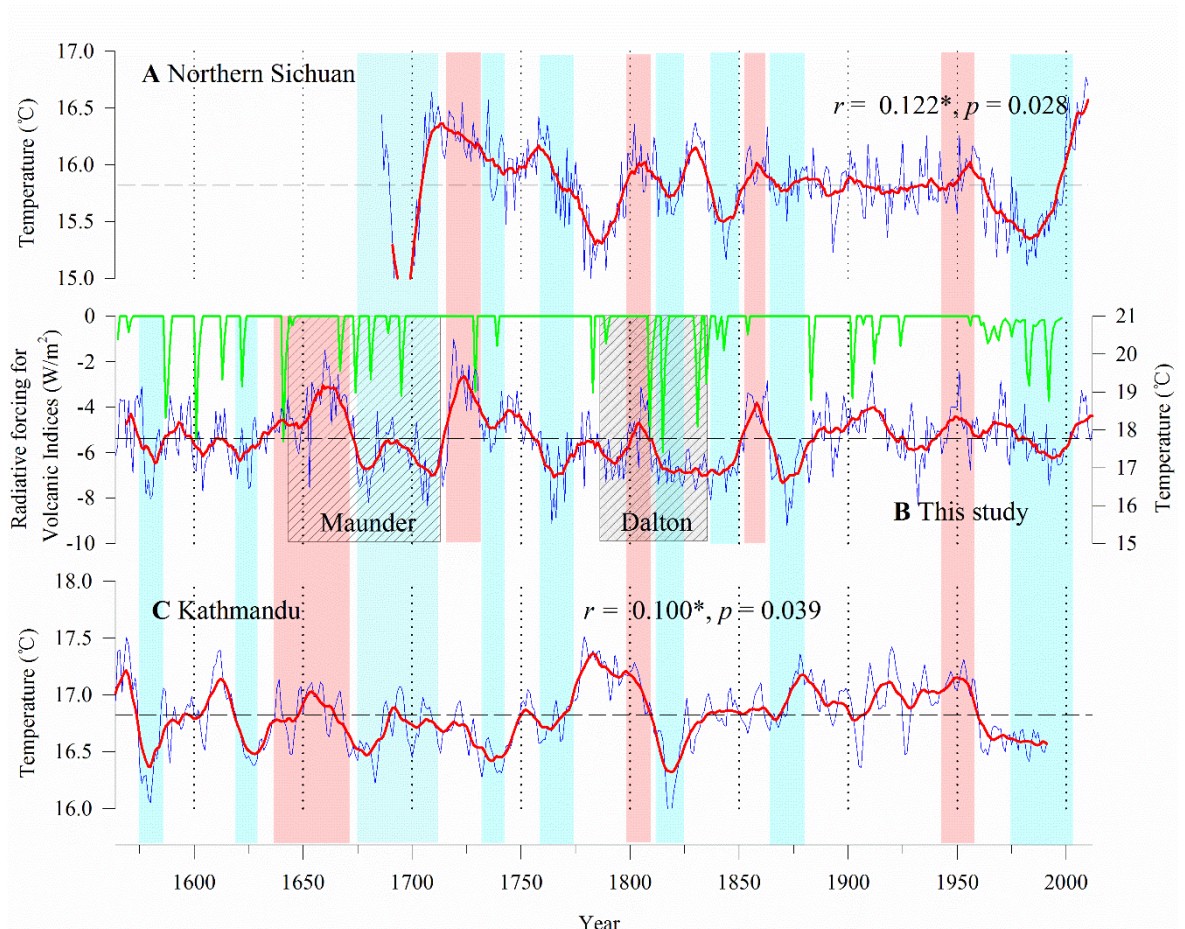

**Fig. 7** Comparison of July-August mean maximum temperature reconstruction of NWSP in this study with other tree-ring based temperature reconstructions. A, July-August mean temperature in northwest Sichun (Xiao et al., 2015); B, July-August mean maximum temperature reconstruction in NWSP (this study); Cyan curve is radiative forcing for ice core volcanic indices series after adjustments (Crowley, 2000) and two boxes filled by oblique parallel lines show the Maunder and Dalton solar minima periods. C, February-June temperature reconstruction in Kathmandu (Cook et al., 2003). All temperature series (blue curve) are smoothed with an 11-year moving average (red curve). Light blue (cold) and red (warm) shading are low and high temperature zones with good agreements in these temperature series, respectively.

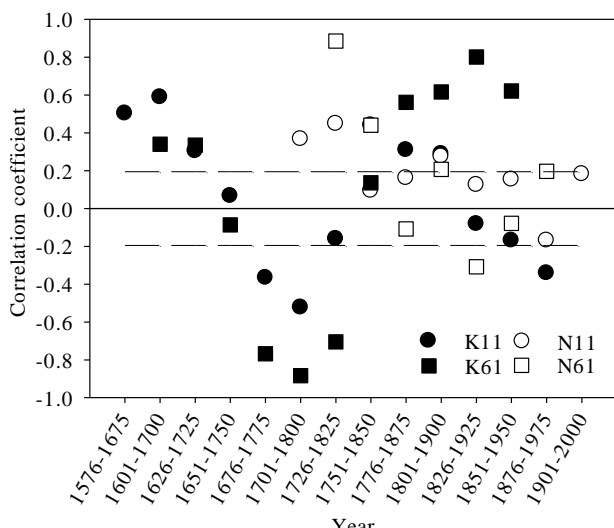

**Fig. 8** Moving correlation function coefficients calculated between our reconstructed temperatures and other temperature reconstructions in decadal (running 11-yr mean) and multidecadal variations (running 61-yr mean). N: July-August mean temperature in northwest Sichun (Xiao et al., 2015), K: February-June temperature reconstruction in Kathmandu (Cook et al., 2003). Period: 1576–2000; moving window: 100 years; overlapping 25 years.

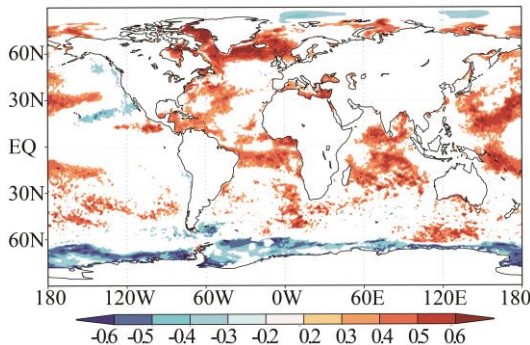

**Fig. 9** Spatial correlations between estimated temperatures and monthly OISSTs at the global scale. The spatial correlation were carried out for months (July-August) covering a time span from 1982 to 2012.

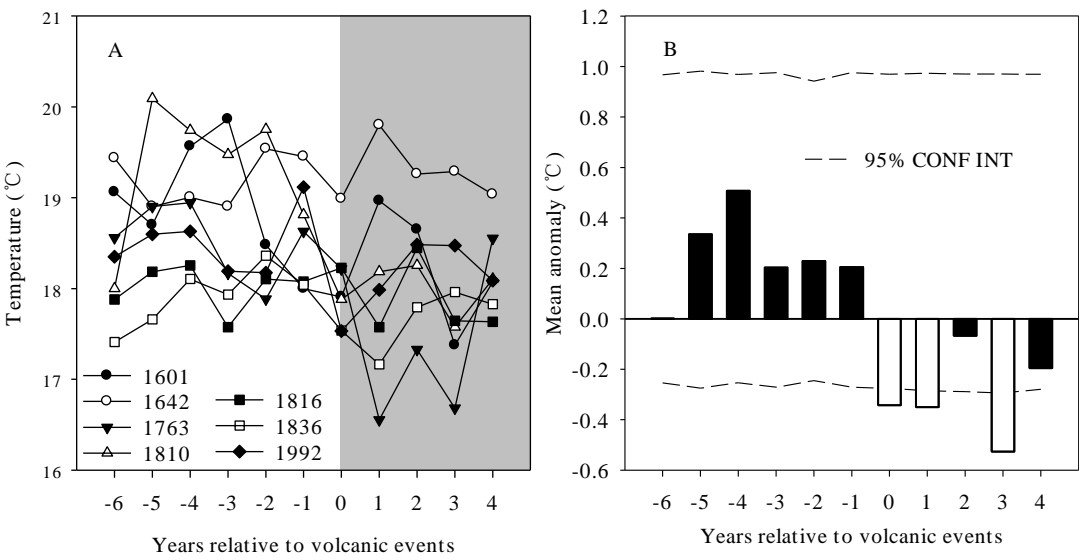

**Fig. 10** The influence of volcanic forcing on temperature variations. Temperature change around the volcanic eruptions (A) and results of the superposed epoch analysis (SEA) for volcanic cooling (B).