# Peer review of "A 368-year maximum temperature reconstruction based on tree ring data in northwest Sichuan Plateau (NWSP), China"

_Climate of the Past, 2016_

## Referee Comment (RC1) · Anonymous Referee #1 · 29 Feb 2016

This manuscript presented a 368-year temperature reconstruction based on tree-ring record in northeast edge of Tibetan Plateau. It is a necessary supplement of past climate proxy records in this area, especially for the monthly mean maximum temperature reconstruction and its implication for rapid warming in recent at this region. Overall this manuscript is well-written, the work seems to be of high quality and is appropriate for Climate of the Past. Therefore, I would recommend this manuscript for publication in this Journal after the following issues are addressed.

(1) In the past 368 years, you identified seven short cold periods and three long warm periods (two long and one short). Could you explain why it appeared like this?

(2) You detected different significant periodicities of temperature variations in the past

368 years. Only for this, you thought the temperature variations could be driven by ENSO, PDO, AMO and solar activity, which may not accurate. Please give more evidences if possible.

(3) References in text of the manuscript should be listed in chronological order.

(4) It would be better if this manuscript is fluent by a native speaker again.

Other detailed comments: Line 11, Page 1: replace "for" with "in" Line 12, Page 1: move "base on a chronology of tree-ring widths over the period 1646-2013" to the end of "variability". Line 15, Page 1: replace "between" with "in", and "in our" with "from this" Line 17, page 1: replace "has" with "had", and add "The" before "20th" Line 21, page 1: add "in this region" at the end of "level" Line 25, page 1: replace "the Tibetan Plateau and the Sichuan Basin" with "Tibetan Plateau and Sichuan Basin" Line 26, page1: delete the first "the" Line 29, page 1: replace "of NWSP" with "in NWSP", delete the last "the" Line 30, page 1: delete the first "the" Line 3, page 2: replace "of NWSP" with "in NWSP" Line 4, page 2: delete the first "the" Line 6, page 2: replace "from NWSP" with "in NWSP" Line 14, page 2: replace "were during a single month" with "reconstructed temperature in a single month" Line 15, page 2: delete "most of them" Line 18, page 2: replace "already has" with "has already" Line 19, page 2: delete "the" before "minimum temperature" and "Northern Hemisphere" Line 22, page 2: delete "the" before "mean temperature" Line 23, page 2: add "Therefore, " before "the aims" Line 24, page 2: add "temperature" before "variations" Line 25, page 2: replace "the new" with "this", and "for" with "in" Line 26, page 2: replace "driver" with "driving", and add "in this area" at the end of this sentence. Line 29, page 2: delete "the" and "forest" Line 31, page 2: delete "type" and "the" Line 4, page 3: delete the space before the first "%" Line 5, page 3: delete "the" before "valley" Line 4, page 3: delete "above sea level (" and ")." Line 17, page 3: replace "standard" with "traditional" Line 18, page 3: delete "the" before "laboratory" Line 19, page 3: delete "the" before "annual" Line 29, page 3: delete "the" before "composite", and add "by" before "estimating" Line 7, page 4: replace "for our" with "in this" Line 8, page 4: delete "the" before "weather" Line 9,

page 4: replace "are the" with "were", and delete "the" before "tree-ring" Line 16, page 4: delete "the" before "tree-ring" Line 20, page 4: replace "of two" with "in two" Line 26, page 4: delete "test" before "(RE)" Line 29, page 4: replace "from" with "in" Line 6, page 5: replace "using SPSS" with "through SPSS", and add a software reference at the end of this sentence. Line 7, page 5: replace "the chronology of Chali" with "Chali chronology", and delete "the" before "Hongyuan" Line 18, page 5: replace "tried to reconstruct" with "reconstructed" Line 19, page 5: add "(TRI)" at the end of "tree-ring index" Line 30, page 5: replace "for" with "in" Line 11, page 6: replace "19.58" with "19.59" Line 2, page 7: delete "was also found" Line 3, page 7: replace "show" with "showed" Line 5, page 7: add "," before "and", replace "has been" with "had been", replace "; most" with ". Most" Line 7, page 7: replace "are " with "were" Line 9, page 7: replace "the growth of spruce trees in treelines of our study" with "the spruce growth at timberline in this area" Line 14-15, page 7: delete ", although late wood would have formed and cambial activity would have stopped" Line 18, page 7: replace "depends" with "depended" Line 20, page 7: replace "has" with "had" Line 24, page 7: replace "tree line has" with "timberline had" Line 26, page 7: replace "explain" with "explained" Line 27, page 7: replace "the tree line" with "timberline" Line 1, page 8: replace "of" with "in" Line 2, page 8: replace "for" with "in" Line 5, page 8: replace "for" with "in", replace "again" with "also" Line 23, page 8: replace "of" with "in" Line 28, page 8: replace "of" with "in" Line 29, page 8: delete "as has been reported" Line 33, page 8: replace "has" with "had" Line 4, page 9: replace "over the" with "in" and replace "detected over the" with "detected in" Line 11, page 9: delete "This result coincides with the fact that", and replace "could be" with "had been" Line 12, page 9: replace "over" with "in" Line 18, page 9: replace "of NWSP" with "in NWSP" Line 23, page 9: replace "evidenced" with "confirmed" Line 27, page 9: replace "for the Eastern" with "in Eastern" Line 28, page 9: replace "of northwest" with "in northwest" Line 1, page 10: replace "of NWSP" with "in NWSP" Line 5, page 10: replace "of NWSP" with "in NWSP" Line 6, page 10: add "temperature" before "variance" Line 10, page 10: add "The" before "20th" Line 1, page 15: replace "on the two" with "of two" Line 7, Table 3, page 16: replace "-0.288*" with

"-0.28**" Line 16, page 17: replace "for" with "in" Line 1, page 18: replace "timberline Picea purpurea in Chali. A, Standard chronology and changing sample size over" with "Picea purpurea at timberline in Chali. A, Standard chronology and sample size changing with" Line 6, page 18: delete "mean" before "total" Line 7, page 18: replace "gray" with "black" Line 2, page 19: replace "for NWSP" with "in NWSP" Line 7-8, page 19: replace "for NWSP" with "in NWSP" Line 1, page 20: is "1645-2012" replaced with "1665-2012"? Line 2, page 20: delete "and 99%" Line 2, page 21: replace "of NWSP" with "in NWSP" Line 3, page 21: replace "A" with "(A)", replace "B" with "(B)" Line 4, page 21: replace "for NWSP" with "in NWSP", "C" with "(C)", and "for Kathmandu" with "in Kathmandu"

---

## Referee Comment (RC2) · Anonymous Referee #2 · 7 Mar 2016

The manuscript presents a July-August mean maximum temperature reconstruction for the West Sichuan Plateau covering the last 368 years. The resulting record is described and compared to other tree-ring based temperature reconstructions in the area. The authors comprehensively investigate the climate signal encoded in the tree rings, critically discuss the physiological meaning of the statistical significant climate-growth relationships found and perform a robust tree-ring based climate reconstruction. The results certainly add some new knowledge to the understanding of regional climate variability and makes it a worthy publication in Climate of the Past.

The manuscript will benefit from a last check by a native speaker. However, readability will improve quite a lot following the careful language check done by reviewer 1.

Find below some general and minor comments.

1) The only potential weakness that I see on the paper refers to the chronology replication. Only 16 trees compose the chronology. This makes me wonder if the EPS has been calculated taking into consideration that more than one core per tree have been included in the chronology. It is not the same to calculate the EPS for an "n" of 32 than 16. On the other hand, I wonder if the few samples reaching back 1650 are cores from the same or different trees. How many trees are covering the part of the chronology with the lowest replication?

2) The authors compared the newly developed reconstruction with pre-existing temperature reconstructions. The correlation values obtained between the different reconstructions are not very high, but they are significant due the high number of observations. The interesting point of comparing different reconstructions is more related to medium and low-frequency climate variations than interannual. Hence, I would suggest to filter the reconstructions and discuss the similarities among them in terms of decadal to multidecadal variations. Moving correlations will definitely help to visualize and describe the periods of agreement and disagreement.

3) It would be also interesting to discuss the existence or not of the Little Ice Age in the new reconstruction.

4) The authors discuss the link between solar changes and temperature variations. What about the volcanic forcing? A large number of studies have reported a link between volcanic eruption and temperature changes. I would suggest to include some analysis (such as superposed epoch analysis) in order to complete the discussion including the influence of volcanic forcing on temperature variations.

5) I would also recommend to rewrite the conclusion section. At the moment looks pretty much like a summary of results and lacks the description of the main findings and their implication on a more general context.

[Figure]

Minor comments: Table 3: "-0.288*" should be changed to "-0.29*"

Figure1: The scale of precipitation should be double the scale of temperature.

Figure 3. Labelling each graph with the corresponding climate variable will make the figure easier to interpret at a fist look.

Figure 4: The results of the residual analysis are embedded in the text but it would also be nice to show the plot of the residuals in this figure.

————————————————————

---

## Author Response (AR1)

wangxc-cf@nefu.edu.cn and zydxju@163.com

**Response to Anonymous Referee #1:**

Thank you for your constructive comments on our manuscript, especially for correcting those grammatical mistakes. All comments are very valuable and helpful for revising and improving our MS, as well as the important guiding significance to our researches. We have studied your comments carefully and have made correction.

**Major comments:**

**1.** In the past 368 years, you identified seven short cold periods and three long warm periods (two long and one short). Could you explain why it appeared like this?

**Response:** Comment accepted. Thanks, we have explained this phenomenon. Those cold or warm periods identified from our reconstructions might be driven by solar and volcanic forcing. The occurrence of the cold or warm periods were nearly consistent with the solar and volcanic forcing. For the detailed information, please look up the lines 23-33, page 9.

**2.** You detected different significant periodicities of temperature variations in the past 368 years. Only for this, you thought the temperature variations could be driven by ENSO, PDO, AMO and solar activity, which may not accurate. Please give more evidences if possible.

**Response:** Comment accepted. Thanks, in addition to the MTM analysis, correlation analysis between our reconstruction and those large-scale atmospheric circulations indices also revealed the significantly correlations between our reconstruction and the ENSO, PDO, AMO and solar activity (see Table 3). On this basis, we added the teleconnection analysis between our reconstruction and SSTs. The RLST variables significantly correlated with the SSTs changes, especially the west and equatorial Pacific as well as the north Atlantic Oceans (Fig. 9), which further verified the driving of large scale climate (e.g. ENSO, PDO and AMO) on regional temperature from another angle (for details, please see the lines 11-15, page 11). In addition, we also added the superposed epoch analysis of volcanic eruption and discussed the influence of volcanic forcing on temperature variations (Fig. 10; for details, please see the lines 24-34, pages 11)

**Table 3** Correlation of the smoothed (11-yr moving averaged) temperature reconstruction and other large-scale climate system cycles.

| Month | AMO (n=156) | PDO (n=112) | MEI (n=62) |
|---|---|---|---|
| PMAY | -0.15 | -0.43** | -0.43** |
| PJUN | -0.19* | -0.26** | -0.42** |
| PJUL | -0.19* | -0.27** | -0.32* |
| PAUG | -0.17* | -0.27** | -0.27* |
| PSEP | -0.17* | -0.24* | -0.27* |
| POCT | -0.07 | -0.19* | -0.29* |
| PNOV | -0.05 | -0.21* | -0.26* |
| PDEC | -0.07 | -0.21* | -0.28* |
| JAN | -0.00 | -0.34** | -0.28* |
| FEB | -0.13 | -0.29** | -0.27* |
| MAR | -0.13 | -0.39** | -0.38** |
| APR | -0.13 | -0.39** | -0.41** |
| MAY | -0.18* | -0.46** | -0.43** |
| JUN | -0.22** | -0.31** | -0.42** |
| JUL | -0.22** | -0.30** | -0.31* |
| AUG | -0.21** | -0.30** | -0.28* |

AMO: Atlantic Multidecadal Oscillation; PDO: Pacific Decadal Oscillation; MEI: Multivariate ENSO Index

* $p < 0.05$, ** $p < 0.01$

[Figure]

**Fig. 9** Spatial correlations between estimated temperatures and monthly OISSTs at the global scale. The spatial correlation was carried out for months (July-August) covering a time span from 1982 to 2012.

[Figure]

**Fig. 10** The influence of volcanic forcing on temperature variations. Temperature change around the volcanic eruptions (A) and results of the superposed epoch analysis (SEA) for volcanic cooling (B).

**3.** References in text of the manuscript should be listed in chronological order.

**Response:** Comment accepted. Thanks, we have listed references in text of our manuscript in chronological order.

**4.** It would be better if this manuscript is fluent by a native speaker again.

**Response:** Comment accepted. Thanks, we have invited a specialist (native-English speaker) made a job of language revision, in order to improve the ability of English expression.

**Other detailed comments:** Line 11, Page 1: replace "for" with "in" Line 12, Page 1...... Line 4, page 21: replace "for NWSP" with "in NWSP", "C" with "(C)", and "for Kathmandu" with "in Kathmandu".

**Response:** Comment accepted. Thanks, we have corrected those mistakes in our MS according to your comments. For the detailed information, please look up the new manuscript.

Once again, thank you very much for your comments and suggestions.

Best Regards,

Liangjun Zhu, on behalf of all co-authors

**Response to Anonymous Referee #2:**

Thank you for your constructive comments on our manuscript. Those comments are all valuable and very helpful for revising and improving our paper, as well as the important guiding significance to our researches. We have studied your comments carefully and have made correction.

1. **Comments:** The manuscript will benefit from a last check by a native speaker. However, readability will improve quite a lot following the careful language check done by reviewer 1.

**Response:** Comment accepted. Thanks for this constructive comment. We have invited a specialist (native-English speaker) made a job of language revision, in order to improve the ability of English expression.

**2. General and minor comments:**

1) The only potential weakness that I see on the paper refers to the chronology replication. Only 16 trees compose the chronology. This makes me wonder if the EPS has been calculated taking into consideration that more than one core per tree have been included in the chronology. It is not the same to calculate the EPS for an "n" of 32 than 16. On the other hand, I wonder if the few samples reaching back 1650 are cores from the same or different trees. How many trees are covering the part of the chronology with the lowest replication?

**Response:** Comment accepted. Thanks, we have recalculated the EPS of the chronology. EPS has an overall mean of 0.9, well above the generally accepted 0.85 cutoff except for a brief period in the 1690s when it falls to just above 0.8 (the Rbar decline to 0.37). The lower EPS and Rbar values Ca.1695 (Fig. 2) seem to result from suppressed growth during this cool period in more mature trees and somewhat erratic juvenile growth in the trees entering the chronology about this time (D'Arrigo et al. 2001). To extend the length of the chronology, we thus consider this chronology to be most reliable over the past 368 years (AD 1646-2013), which corresponds to a minimum sample depth of 5 cores (four trees) for our chronology (Fig. 2). In order to avoid confusing the reader, we have added some related description about the replication and EPS of the chronology in the text. Please see lines 9-16, page 4 in the text.

[Figure]

**Fig. 2** The tree-ring chronology of *Picea purpurea* at timberline in Chali. A, Standard chronology and sample size changing with time and B, Rbar and EPS (see text for explanation) plotted for 50-year windows with 25-year overlap for the standard chronology. The *vertical dashed line* represents the cut-off point when the n ≥ 5 and EPS > 0.8. The *horizontal red dashed line* represents the theoretical 0.8 threshold.

2) The authors compared the newly developed reconstruction with pre-existing temperature reconstructions. The correlation values obtained between the different reconstructions are not very high, but they are significant due the high number of observations. The interesting point of comparing different reconstructions is more related to medium and low-frequency climate variations than interannual. Hence, I would suggest to filter the reconstructions and discuss the similarities among them in terms of decadal to multidecadal variations. Moving correlations will definitely help to visualize and describe the periods of agreement and disagreement.

**Response:** Comment accepted. Thanks, we have filtered our and other reconstructions with 11- (medium-frequency) and 61-yr (low-frequency) moving averaged, and have visually described the similarities between them using the moving correlations. Our reconstruction temperatures on the whole showed consistency with the near reconstruction in decadal (11-yr moving averaged) variations and the far away reconstruction multidecadal (61-yr moving averaged) variations, although some potentially differences remain (see lines 9-16, page 9 in the text; Fig. 8). However, those decadal or multidecadal differences between them might be due to differences in season, species and the different standardization approach applied on tree-ring data as well as those inherent differences in climate variables and its driven mechanisms (see lines 16-23, page 5 in the text).

[Figure]

**Fig. 8** Moving correlation coefficients calculated between our reconstructed temperatures and other temperature reconstructions in decadal (running 11-yr mean) and multidecadal variations (running 61-yr mean). N: July-August mean temperature in northwest Sichun (Xiao et al., 2015), K: February-June temperature reconstruction in Kathmandu (Cook et al., 2003). Period: 1576–2000; moving window: 100 years; overlapping 25 years.

3) It would be also interesting to discuss the existence or not of the Little Ice Age in the new reconstruction.

**Response:** Comment accepted. Thanks, we have added the discussion of the existence or not of the Little Ice Age in our reconstruction. The Little Ice Age (LIA) climate obviously exist in our reconstruction and end with the climatic abrupt change of the end of 19th century. Changes in temperature as well as the frequency of low temperature events both show that the 18th and 20th centuries were mainly in warm and the 17th and 19th centuries were in cold, which were in agreement with other study of the LIA. For the detailed information, please look up the lines 2-7, page 10 of the discussion section.

4) The authors discuss the link between solar changes and temperature variations. What about the volcanic forcing? A large number of studies have reported a link between volcanic eruption and temperature changes. I would suggest to include some analysis (such as superposed epoch analysis) in order to complete the discussion including the influence of volcanic forcing on temperature variations.

**Response:** Comment accepted. Thank you for this constructive comment. We have added some analysis between volcanic eruptions and temperature variations and discussed the influence of volcanic forcing on temperature variations. Results show that volcanic forcing has an important influence on temperature variations, which has found in other dendroclimatological studies in the Tibetan Plateau, Central Asia, Europe as well as the Northern hemisphere (Fischer et al., 2007;Liang et al., 2008;Breitenmoser et al., 2013;Davi et al., 2015;Stoffel et al., 2015). On one hand the radiative forcing for volcanic indices series (Crowley, 2000) presents a high consistency with our temperature reconstruction (see Fig. 7B). On the other hand most

of volcanic eruptions events lead a temperature drop in the year and subsequent years and the result of SEA also shows significant cooling occurred during the year of the event as well as the following years (see Fig. 10). For the detailed information, please look up the lines 24-34, page 11 of the discussion section.

[Figure]

**Fig. 7** Comparison of July-August mean maximum temperature reconstruction of NWSP in this study with other tree-ring based temperature reconstructions. A, July-August mean temperature in northwest Sichun (Xiao et al., 2015); B, July-August mean maximum temperature reconstruction in NWSP (this study); Cyan curve is radiative forcing for ice core volcanic indices series after adjustments (Crowley, 2000) and two boxes filled by oblique parallel lines show the Maunder and Dalton solar minima periods. C, February-June temperature reconstruction in Kathmandu (Cook et al., 2003). All temperature series (blue curve) are smoothed with an 11-year moving average (red curve). Light blue (cold) and red (warm) shading are low and high temperature zones with good agreements in these temperature series, respectively.

[Figure]

**Fig. 10** The influence of volcanic forcing on temperature variations. Temperature change around the volcanic eruptions (A) and results of the superposed epoch analysis (SEA) for volcanic cooling (B).

5  5) I would also recommend to rewrite the conclusion section. At the moment looks pretty much like a summary of results and lacks the description of the main findings and their implication on a more general context.

**Response:** Comment accepted. Thanks, we have rewrite the conclusion section. Following is the new conclusion:

In this study, a high resolution tree-ring chronology was used to reconstruct RLST in NWSP from 1645 to 2012. The model of our reconstruction explains 37.1 % of the temperature variance. Spatial correlations with gridded land mean

10  maximum temperature data and comparison with other nearby temperature reconstructions further revealed that the RLST has high spatial representativeness, although there have some temporal and spatial differences in the low-frequency changes of temperature. Seven major cold periods and three major warm periods were identified from this reconstruction, which might be linked with volcanic and solar activities. The 18th and 20th centuries were mainly in warm with a lower frequency of low temperature events, while the 17th and 19th century were cold with a higher frequency of extreme low temperature events.

15  Significant periodicities at 2-2.3, 2.5, 3-3.1, 4.6-4.7, 5.6, 25-32 and 53-93 years exist in our reconstruction. The 20th century rapid warming wasn't obvious in our RLST reconstruction, which implies that mean maximum temperature, as a unique temperature indicator, might play an important and different role in global change. Moreover, the Little Ice Age (LIA) climate can also be well represented and obviously end with climatic amelioration at the end of the 19th century. Overall, the RLST variability in NWSP might be associated with global land-sea atmospheric circulation (e.g. ENSO, PDO or AMO) as well as

20  solar and volcanic forcing.

**3. Minor comments:**

1) We have followed your suggestion to modify the "-0.288*" to "-0.29*" in Table 3.

**Response:** Comment accepted. Thanks, we have replace "-0.288*" with "-0.29*" in Table 3.

2) Figure 1: The scale of precipitation should be double the scale of temperature.

5    **Response:** Comment accepted. Thanks, the scale of precipitation has been adjusted to double the scale of temperature. Details are as follows:

[Figure]

**Fig. 1** Monthly variation of total precipitations (bars), mean maximum temperature (line with squares), mean temperature (line with circles), and mean minimum temperature (line with triangles) in Hongyuan (filled with white) and Aba (filled with black)

10   meteorological stations, calculated for the period of 1961–2013 and 1955–1990, respectively.

3) Figure 3. Labelling each graph with the corresponding climate variable will make the figure easier to interpret at a fist look.

**Response:** Comment accepted. Thanks, we have labelled each graph with the corresponding climate variable. Details are as follows:

[Figure]

**Fig. 3** Correlation analyses between the Chali tree-ring chronology and meteorological climate data of Hongyuan and Aba including total precipitation (A, E), mean temperature (B, F), minimum temperature(C, G), and maximum temperature (D, H). The black-filled bars represent significant effects at 95 % significant levels.

4) Figure 4: The results of the residual analysis are embedded in the text but it would also be nice to show the plot of the residuals in this figure.

**Response:** Comment accepted. Thanks, we have added the residual results into Figure 4. Details are as follows:

[Figure]

**Fig. 4** Temperature reconstruction in NWSP, China. A, The comparison of observed (pink line) and reconstructed (blue line) regional Jul-Aug mean maximum temperature during the calibration period 1955-2012; B, Shapiro–Wilk residuals normality test (SW); C, Durbin-Watson test for residuals autocorrelation (DW); D, Annual and 11-year smoothed regional Jul-Aug mean maximum temperature reconstruction (blue and tick red lines, respectively). Superimposed is the 11-year smoothed instrumental record (pink line).

Once again, thank you very much for your comments and suggestions.

Best Regards,

Liangjun Zhu, on behalf of all co-authors

[revised manuscript text omitted]